# Anti-Cancer Effects of Artesunate in Human 3D Tumor Models of Different Complexity

**DOI:** 10.3390/ijms24097844

**Published:** 2023-04-25

**Authors:** Marlene Niederreiter, Julia Klein, Kerstin Arndt, Jens Werner, Barbara Mayer

**Affiliations:** 1Department of General, Visceral, and Transplant Surgery, Ludwig-Maximilians-University Munich, Marchioninistraße 15, 81377 Munich, Germany; marlene.niederreiter@med.uni-muenchen.de (M.N.); jens.werner@med.uni-muenchen.de (J.W.); 2German Cancer Consortium (DKTK), Partner Site Munich, Pettenkoferstraße 8a, 80336 Munich, Germany; 3SpheroTec GmbH, Am Klopferspitz 19, 82152 Martinsried, Germany

**Keywords:** artemisinin-derivatives, Artesunate, 3D cancer model, cancer spheroid model, neoplasia

## Abstract

The anti-malaria drug Artesunate (ART) shows strong anti-cancer effects in vitro; however, it shows only marginal treatment results in clinical cancer studies. In this study, ART was tested in preclinical 3D cancer models of increasing complexity using clinically relevant peak plasma concentrations to obtain further information for translation into clinical use. ART reduced cell viability in HCT-116 and HT-29 derived cancer spheroids (*p* < 0.001). HCT-116 spheroids responded dose-dependently, while HT-29 spheroids were affected more strongly by ART than by cytostatics (*p* < 0.001). HCT-116 spheroids were chemo-sensitized by ART (*p* < 0.001). In patient-derived cancer spheroids (PDCS), ART led to inhibition of cell viability in 84.62% of the 39 samples tested, with a mean inhibitory effect of 13.87%. Viability reduction of ART was 2-fold weaker than cytostatic monotherapies (*p* = 0.028). Meanwhile, tumor-stimulation of up to 16.30% was observed in six (15.38%) PDCS-models. In 15 PDCS samples, ART modulated chemotherapies in combined testing, eight of which showed chemo-stimulation (maximum of 36.90%) and seven chemo-inhibition (up to 16.95%). These results demonstrate that ART’s anti-cancer efficacy depends on the complexity of the tumor model used. This emphasizes that cancer treatment with ART should be evaluated before treatment of the individual patient to ensure its benefits and prevent unwanted effects.

## 1. Introduction

The use of complementary and alternative medicine (CAM) by cancer patients is growing rapidly [1,2,3,4,5]. This trend is also reported for Natural Products [6,7], one of the main CAM categories defined by the National Center for Complementary and Alternative Medicine [8]. The main reasons why cancer patients decide to use natural products include improvement of tumor response to standard cancer therapy, mitigation of therapy related side effects, boosting the immune system and increasing quality of life [9,10].

One of the best-known natural substances in Traditional Chinese Medicine is artemisinin, extracted from the herb *Artemesia annua* [11]. Artesunate (ART), which is a semi-synthetic, hydrophilic derivative of artemisinin, is among the most potent and well-tolerated drugs used to fight malaria [12,13,14]. ART has been repurposed as an anti-cancer drug. Numerous research studies with cancer cell lines in vitro and in animals have shown that ART inhibits multiple signaling pathways involved in tumor promotion. This includes suppression of tumor cell proliferation, cell cycle arresting, induction of different forms of tumor cell death and inhibition of angiogenesis, invasion and metastasis [15,16,17,18,19,20,21,22,23].

In contrast to this extensive research, there are very few Phase I anti-neoplasia trials published in English that have investigated artemisinin-type drugs in small cohorts [24,25,26,27,28,29] (summarized in Appendix A). ART was well tolerated after intravenous injection [24], oral and intravaginal applications [25,26,27,28,29]. Toxicities were primarily low-grade and often involved fatigue, nausea and anemia [25,26,27,28]. Although tumor responses included symptom control, stable disease [24,26,27], histologic regression [28] and survival prolongation in a few patients [29], as well as changes in the progression-related tumor biomarker profile [25,29] in others, general anti-tumor effects have only been marginal thus far. The discrepancy between experimental and clinical data highlights the need for patient-centered research to clarify the impact of artemisinin derivatives in cancer treatment [30]. Therefore, the therapeutic effect of ART as a single agent and in combination with guideline-recommended chemotherapies was analyzed in the present study using clinically relevant peak plasma concentrations in human 3D cancer models of increasing complexity. Specifically, the spheroid model, derived from cancer cell lines, is well known to mimic homotypic tumor-cell-interactions and the biochemical tumor microenvironment, such as hypoxia and acidosis [31,32,33]. The patient-derived cancer spheroid model, which is generated directly from patients’ tumor tissue, additively reflects heterotypic cell-cell interactions and was found predictive for the treatment response of individual cancer patients [34,35,36]. ART testing in complex biological models using clinically relevant dosing could provide important information for translation into clinical use.

## 2. Results

### 2.1. Anti-Cancer Effects of Artesunate in Cell-Line Derived Cancer Spheroids

#### 2.1.1. Artesunate Induces Morphological Changes in HCT-116 and HT-29 Derived Cancer Spheroids

Treatment of HCT-116 spheroids with Artesunate (ART) resulted in a cloudy spheroid structure with a loose center and periphery that broke into cell clusters and single cells (Figure 1B). HT-29 spheroids also acquired a loose morphology after ART treatment (Figure 1D). In comparison, ART’s solvent control (SC), consisting of water and sodium hydrogen carbonate (8.4%), showed no morphological changes (Figure 1A,C).

#### 2.1.2. Artesunate Reduces Viability in HCT-116 and HT-29 Derived Cancer Spheroids

As a single agent, ART moderately affected the viability of HCT-116 spheroids in a dose-dependent manner (Figure 2A), as ART treatment with a peak plasma concentration (PPC) of 15 µg/mL resulted in a significantly higher decrease in viable cells (35.92 ± 1.36%) when compared to a PPC of 0.74 µg/mL (24.45 ± 3.69%, *p* < 0.001). In comparison to ART monotherapy, three guideline-recommended chemotherapies for colorectal cancer were tested: 5-fluorouracil (5FU), 5FU + Oxalipaltin (FO) and 5FU + Irinotecan (FI) [37,38]. Treatment with each cytostatic showed a strong anticancer effect that was superior to the therapy with ART (5FU versus ART, PPC 15 µg/mL, *p* < 0.001; FO versus ART, PPC 15 µg/mL, *p* = 0.005; FI versus ART, PPC 15 µg/mL, *p* < 0.001). 

In contrast to HCT-116 spheroids, ART treatment in HT-29 spheroids showed a strong, dose-independent anti-cancer effect (Figure 2B). Remarkably, in HT-29 spheroids, ART monotherapy decreased cell viability significantly more strongly than each tested chemotherapy (all *p* < 0.001). The solvent control (SC) of ART did not display noteworthy anti-tumor properties (HCT-116 cancer spheroids: mean cell viability of −4.44 ± 3.97%; HT-29 cancer spheroids: mean cell viability of −1.77 ± 5.76%).

#### 2.1.3. Artesunate Shows Chemo-Sensitizing Properties in HCT-116 Derived Cancer Spheroids

The addition of Artesunate (ART) resulted in the chemo-sensitization of standard chemotherapies in HCT-116 spheroids (Figure 3). At PPCs of 0.74 µg/mL (Figure 3A) and 15 µg/mL (Figure 3B), ART increased the anti-cancer effects of 5-Fluorouracil (5FU), 5FU + Oxaliplatin (FO) and 5FU + Irinotecan (FI) (all *p* < 0.001).

As described earlier, ART monotherapy showed a significantly stronger decrease in cell viability than each tested chemotherapeutic in HT-29 spheroids. At a dose of 0.74 µg/mL, combination therapy of ART and FI did not show any effects beyond ART alone. However, the combination with 5FU and FO was slightly, but significantly, stronger than ART monotherapy (*p* < 0.001 and *p* = 0.004) (Figure 4A).

In contrast, the combination therapy of each cytostatic with ART at 15 µg/mL was just as effective as ART alone (Figure 4B).

### 2.2. Anti-Cancer Effects of Artesunate in Patient-Derived Cancer Spheroids

#### 2.2.1. Patient Characteristics

Tumor samples were collected from 39 patients to generate patient-derived cancer spheroids (PDCS). Most cancer patients were female (25 of 39 patients, 64.10%) and diagnosed in a progressed tumor stage (27 of 39 patients, 69.23%). After gynecological tumors (17 of 39 patients, 43.59%), rare tumors (11 of 39 patients, 28.21%) formed the second largest group; these included gastrointestinal stromal tumor (GIST), liposarcoma, Bowen’s disease (squamous cell carcinoma in situ), pleural mesothelioma, glioblastoma, malignant solitary fibrous tumor of the pleura, hepatoblastoma, neuroendocrine tumor (NET), vulvar cancer and peritoneal mesothelioma (Table 1).

#### 2.2.2. Artesunate Induces Morphological Changes in Patient-Derived Cancer Spheroids

Patient-derived cancer spheroids (PDCS) treated with 0.74 µg/mL ART displayed morphological changes (Figure 5). In comparison to its solvent control (Figure 5A,C), ART led to more compact spheroids and a microscopically visible decrease of spheroid diameter. Further on, ART treated PDCS showed multiple cells of decreased size in the periphery of the spheroid and frayed edges (Figure 5B,D).

#### 2.2.3. Artesunate Modulates Cell Viability in Patient-Derived Cancer Spheroids and PBMC

Treatment of patient-derived cancer spheroids with ART inhibited cell viability in 33 of 39 (84.62%) samples. The mean inhibitory effect of ART amounted to 13.87% (Figure 6). A comparison between ART and the standard chemotherapies revealed the effect of ART to be significantly inferior to those of cytostatics tested (ART monotherapy, mean 13.87%, versus CTx monotherapy, mean 28.15%, *p* = 0.028; ART versus doublet CTx, mean 48.93%, *p* < 0.001; ART versus triplet CTx, mean 58.62%, *p* < 0.001, Figure 6). Meanwhile, in six (15.38%) patients, ART stimulated tumor spheroid viability. No correlation was found between ART efficacy and different parameters of the patient cohort (age, gender, tumor status and tumor entity), as described in Appendix A.

Treatment of peripheral blood mononuclear cells (PBMC) with ART resulted in inhibiting effects overall (mean 5.76%). PBMC viability was reduced by up to 32.91% in 12 samples (75.00%) (Figure 6). Additionally, at a PPC of 0.74µg/mL, ART induced hemolysis in four samples (Figure 7). In contrast, ART treatment led to PBMC stimulation in four out of 16 (25.00%) cases, ranging from 4.7% to 17.84%. In comparison, treatment of PBMC with chemotherapeutics led to a significantly higher inhibition of cell viability than ART (ART vs. CTx monotherapy, *p* = 0.06, Figure 6).

The solvent control of ART revealed marginal effects on both patient-derived cancer spheroids (−2.65 ± 16.05%) and PBMC (mean −3.84 ± 7.99%).

#### 2.2.4. Artesunate Modulates Chemotherapy in Patient-Derived Cancer Spheroids

In total, 15 guideline- and physician-recommended combinations of ART and cytostatics were tested according to tumor entity. In eight samples, ART led to additional enhanced cytostatic efficacy with a mean value of 16.56%, ranging from 3.47% to 36.90% (Figure 8A). Conversely, chemo-reducing effects in combination with ART were noticed in seven cases (mean chemotherapy reduction of 7.23% with a range of 1.42% to 16.95%; Figure 8B).

#### 2.2.5. Anti-Cancer Effects of Artesunate Depend on Individual Tumor Samples

The highest effect of ART monotherapy (54.48%) was observed in recurrent cervical cancer spheroids and was more effective than all chemotherapeutic agents comparatively tested for this individual patient. Nevertheless, ART treatment was not significantly better than Cisplatin, the most effective chemotherapeutic (Figure 9, patient 1). One other patient showed similar results to patient 1, with ART being the strongest treatment tested. Furthermore, ART was equally effective as the tested cytostatics in three other samples (Figure 9, patient 3). As described in Section 2.2.3, ART also led to stimulation of cell viability. The stimulatory maximum (16.30%) was observed in PDCS deriving from a recurrent liposarcoma (Figure 9, patient 2).

Regarding chemo-modulatory activities, treatment with ART caused both chemo-stimulatory and chemo-inhibitory effects. Patient 4 in Figure 10 displayed a significant chemo-enhancement (1.33-fold) through combination with ART in comparison to temozolomide-monotherapy (*p* < 0.001). In addition, PDCS derived from a metastatic gastric cancer patient responded significantly more strongly to a combination of FI and ART in comparison to FI alone (Figure 10, patient 5; *p* < 0.001). Patient 6 is an example of the reduction of chemo-sensitivity (1.65-fold) of Carboplatin and Paclitaxel(Figure 10) through combination with ART.

## 3. Discussion

The anti-malaria drug Artesunate (ART) was repurposed as an anti-cancer drug based primarily on promising results obtained in two-dimensional (2D)-cell culture models. However, in the clinical setting, ART-related benefits for cancer patients are low. This discrepancy suggests the limited predictive power of standard cell culture models for ART’s anti-cancer activity. Therefore, the present study investigated the anti-cancer effects of ART in 3D-cell culture models of increasing complexity using clinically relevant peak plasma concentrations. It has been well described that 3D cell culture models recapitulate a variety of tumor tissue characteristics, including cell-cell interactions, cell-matrix interactions, the tumor microenvironment and drug resistance [31,32,33].

### 3.1. Artesunate Induced Substantial Anti-Tumor Effects in Homotypic Cell Culture Models

In ART science, the use of 3D-cancer cell models is scarce. In gynecological tumor entities, exposure of MCF-7 breast cancer spheroids [39] and HEY ovarian cancer spheroids [40] to ART resulted in dose dependent changes of spheroid morphology and reduction of cancer cell viability. In gastrointestinal cancers, susceptibility to Dihydroartemisinin (DHA), another artemisinin-derivative, was shown in HCT-116 and DLD-1 colorectal cancer spheroids. In addition, DHA sensitized CRC spheroids to TRAIL-induced apoptosis [41]. The novel artemisinin derivative FO8643 was tested in HUH-7 hepatocellular carcinoma spheroids and inhibited cell migration dose- and time-dependently [42]. In line with this data, the present study found that ART treatment of HT-29 spheroids resulted in a strong, dose-independent reduction of cell viability, while the efficacy of ART in HCT-116 spheroids was moderate and dose-dependent. Further on, ART sensitized HCT-116 spheroids towards standard chemotherapeutics; this is similar to previous reports for different artemisinin-derivatives and a number of cancer cell lines in 2D-cell cultures [43,44,45,46].

### 3.2. Artesunate Induced Heterogeneous Anti-Tumor Effects in Heterotypic Cancer Spheroid Models

While cell line-based cancer spheroids developed moderate to strong responses towards ART, patient-derived cancer spheroids (PDCS) responded in a more heterogeneous manner. Although ART showed, on average, low inhibitory effects in PDCS, these still ranged from moderate suppression of cell viability to stimulatory properties, a circumstance not encountered in any cell line based experiment.

Heterogeneous responses, ranging from comparatively sensitive to quite resistant towards ART treatment, were also observed in patient-derived ovarian tumor organoids [47]. Similarly, patient-derived high-grade glioma spheroids revealed variable responses pattern to DHA [23]. The data obtained in patient-derived tumor models strongly suggest including advanced 3D modelling for informative drug testing and successful translation into clinical application.

In fact, in clinical trials, artemisinin derivatives exerted favorable effects in cancer patients, although these were only minor [24,25,26,27,28,29]. Therefore, several strategies have been developed to enhance the anti-tumor effects of ART treatment. These include the design of new artemisinin derivatives, such as Artesunate-loaded nanoparticles [48], the conjugation of Artesunate with aptamers [49] and chemically modified artemisinin derivatives [42,50]. Another possibility lies in the combination of ART with various established drugs, including the angiotensin converting enzyme (ACE) inhibitor Captopril [51], the Ras inhibitor Farnesylthiosalicylic acid [52] and the heme precursor 5-Aminolevulinic acid (5-ALA) [23].

### 3.3. Artesunate Induced Unwanted Effects 

In contrast to the anti-tumorous effects of ART, cell viability in a minor fraction of PDCS samples was stimulated by ART treatment. In addition, ART in combination with cytostatic drugs was able to diminish chemotherapeutic outcome, a circumstance with possibly severe clinical consequences. Potential interactions of ART with other drugs might be associated with the enrichment of cancer-stem-like cells [53], the modulation of drug transporter systems [54,55] and, in patients, with the interaction of cytochrome P450 (CYP) enzymes [56]. Further unwanted effects induced by ART were observed on peri-pheral blood mononuclear cells (PBMC). PBMC were inhibited in cell viability and responded partly with hemolysis towards ART treatment, especially at higher concentrations. Post-Artesunate delayed hemolysis is a side effect encountered in approximately 15% of ART-treated malaria patients [12]. ART was also shown to have possible cytotoxic effects in vitro in normal cells. Yin et al. [57] reported increased apoptosis rates and decreased cell viability in normal liver cells; in addition, Li et al. [58] reported similar findings for bronchial epithelial cells that were damaged by ART treatment.

Additional toxicities were observed in several clinical studies of ART in cancer patients. Although most adverse events showed a minor grade of severity, they still affected some body systems (blood and lymphatics) with a drop in reticulocytes, neutropenia and anemia [24,25,26,27], as well as the gastrointestinal tract with nausea, abdominal pain, diarrhea and vomiting [24,25,26,27,29] (Appendix A). 

Cancer patients using complementary substances to fight cancer have high expectations of the use of complementary and alternative medicine (CAM) and often are not aware of its potential risks [59]. Considering this, it becomes clear that physicians, pharmacists and alternative therapists are obligated to inform their cancer patients sufficiently about the potential benefits and risks of treatment with ART and other complementary substances [60]. Conversely, patients need to properly inform their physicians about the intake of natural substances. Pre-therapeutic testing of artemisinin derivatives in the patient-derived cancer spheroid model will support treatment optimization for the individual cancer patient.

## 4. Materials and Methods

### 4.1. Cancer Cell Lines and Cell Line-Derived Cancer Spheroids

Human colorectal cancer cell lines HCT-116 and HT-29 were selected because of their reported sensitivity towards Artesunate [61,62] and their spheroid formation ability [63,64,65]. Cell lines were purchased from ATCC (HCT-116, ATCC CCL-247; HT-29, ATCC HTB-38) and cultured according to the manufacturer’s instructions. Cell line-derived cancer spheroids were prepared as described previously [63]. Briefly, monolayer cultures with a confluency of at least 90% were detached with 0.05% trypsin/0.53 mM EDTA (Pan Biotech, Aidenbach, Germany). Cell number and viability were determined using the trypan blue exclusion assay. Cell suspensions with a viability of at least 90% were used for spheroid formation. 5 × 10^4^ vital cells were seeded in each well of a low adhesive 96-well microtiter plate and incubated for 48 h under standard culture conditions (37 °C, 5% CO_2_). A single cancer spheroid was obtained in each well. Cancer cell lines were used for a maximum of 10 passages. Mycoplasma assays were performed quarterly.

### 4.2. Cancer Tissues and Patient-Derived Cancer Spheroids

Double-coded samples (n = 39) of different tumor entities and the corresponding clinical data were provided by the Biobank of the Department of General, Visceral and Transplantation Surgery at the Ludwig-Maximilians-University (LMU) Munich and as a part of the SpheroID study with reference number 278/04 of the ethics committee of the LMU Munich. The Biobank operates under the administration of the Human Tissue and Cell Research (HTCR) Foundation. The framework of the HTCR Foundation, which includes obtaining written informed consent from all donors, has been approved by the ethics commission of the Faculty of Medicine at the LMU (approval number 025-12) as well as the Bavarian State Medical Association (approval number 11142) in Germany.

Patient-derived cancer spheroids were prepared directly from fresh tumor tissues without expansion culture as described in detail [34,35,36,63,66]. In brief, a cell suspension was prepared by mechanical and enzymatic digestion of the tumor samples with a Liberase enzyme cocktail according to the manufacturer’s protocol (Roche, Mannheim, Germany). Primary suspension cells were counted and tested for viability using the Trypan blue exclusion test. No depletion or enrichment of a specific cell type was performed. Seeding of 5 × 10^4^ vital cells resulted in a single patient-derived cancer spheroid per 96-well after 48 h.

### 4.3. Whole Blood Samples and Isolation of PBMC

Anticoagulated blood samples were donated by 20 of the 39 cancer patients and provided through the Biobank. Blood samples were diluted with D-PBS (PanBiotech, Aidenbach, Germany) in a 1:1 ratio, followed by BioColl^®^ (Bio&Sell, Feucht, Germany) density gradient centrifugation at 1200 g for 30 min according to standard protocols [67]. Peripheral blood mononuclear cells (PBMC) were visible as interphase. Erythrocytes in the interphase varied depending on the individual patient sample and were not removed. Cells of the interphase were harvested and washed twice with D-PBS. The number and viability of PBMCs was determined using the Trypan blue exclusion test. 5 × 10^4^ vital cells/well were seeded in a 96-well plate.

### 4.4. Treatment and Cell Viability Analysis

Cell line-derived cancer spheroids were treated with ART (Burg-Apotheke, Königstein, Germany) as a single agent in three different clinically relevant peak plasma concentrations (PPCs; Table 2). PPCs were selected in accordance with clinical trial results (Appendix A). As recommended by the guidelines for the treatment of colorectal cancer, HCT-116 and HT-29 derived cancer spheroids were treated with 5-Fluorouracil (5FU) as a single agent and in combination with Oxaliplatin (FO) or Irinotecan (FI), all from the LMU Pharmacy. All concentrations used are summarized in Table 2. 

In PDCS, the effect of ART was compared to one or several mono or combination chemotherapies recommended by the treatment guidelines and the responsible therapists depending on the type of cancer studied. For chemo modulation studies, ART was combined with guideline or physician recommended chemotherapeutics, according to tumor entity in the PDCS-model.

PBMC were treated with the same ART concentrations as described above. However, treatment at higher concentrations (i.e., 3.26 µg/mL and 15 µg/mL) often resulted in hemolysis (Figure 7), whereas this side effect was rare using a PPC of 0.74 µg/mL.

In consequence, ART treatment of the patient-derived cancer spheroids was limited to the concentration of 0.74 µg/mL in both monotherapy and combination therapy. For each treatment, the appropriate solvent control (SC) was considered. The SC for ART was water for injection and sodium hydrogen carbonate 8.4% (manufactured for this study by the Burg-Apotheke, Königstein, Germany).

### 4.5. Statistical Analysis

Results are presented as means ± SD. Statistical analyses for the comparison of Artesunate and cytostatic induced therapeutic effects in cell-line based cancer spheroids were performed with one-way ANOVA and corrected for multiple testing with Bonferroni’s correction. Data based on patient-derived cancer spheroids was analyzed with the Kruskal–Wallis test and corrected with Bonferroni’s correction. Patient cohort data was tested for correlation with therapeutic effects by 2-tailed Fisher’s exact *t*-test. *p*-values lower than 0.05 were considered significant. All calculations were carried out using the statistics program SPSS (version 27) by IBM and presented with GraphPad Prism (version 9). 

## 5. Conclusions

In the patient-derived cancer spheroid model, Artesunate (ART) acts as a medicinal product that can induce both tumor-stimulation and tumor-inhibition, as well as chemo-sensitization and reduction, depending on the individual tumor. Anti-cancer activity of ART in the PDCS-model as a single agent is low, as well as in clinical trials. Enhancement strategies need to be clinically investigated to improve anti-cancer efficacy of ART in cancer patients. Patients need to be informed that unwanted effects may occur due to ART treatment. Complex tumor models are a necessity to select the optimal treatment for the individual cancer patient.

## Figures and Tables

**Figure 1 ijms-24-07844-f001:**
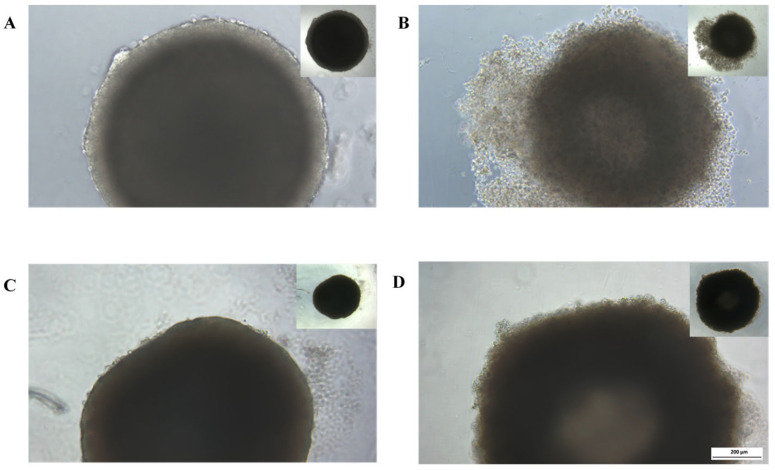
Cell line derived cancer spheroids treated with 15 µg/mL of Artesunate (**B**,**D**) or the cor-responding solvent control (**A**,**C**) for 72 h. (**A**,**B**) HCT-116 derived cancer spheroids. (**C**,**D**) HT-29 derived cancer spheroids. Photographs in the corner show spheroids at 40× magnification, while larger photographs show spheroids at 100× magnification; white bar in the corner shows a length of 200 µm.

**Figure 2 ijms-24-07844-f002:**
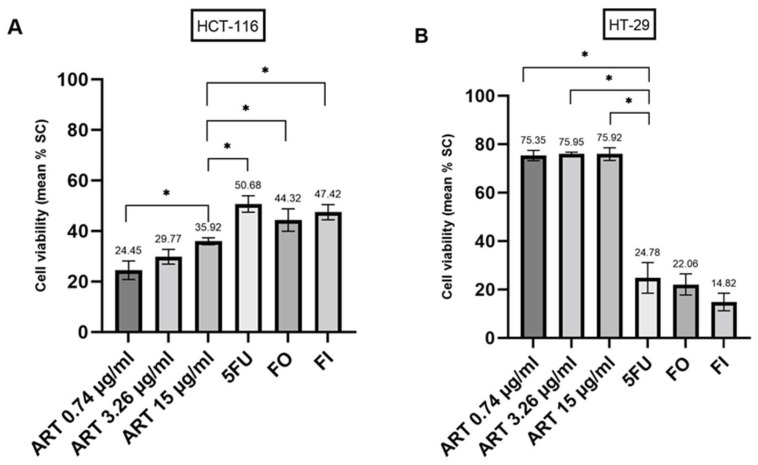
Anti-cancer effects of Artesunate (ART) in different peak plasma concentrations compared to guideline recommended chemotherapies in HCT-116 (**A**) and HT-29 (**B**) derived cancer spheroids. 5FU, 5-Fluorouracil; FO, 5FU+Oxaliplatin; FI, 5FU+Irinotecan; SC, solvent control. * *p* < 0.01.

**Figure 3 ijms-24-07844-f003:**
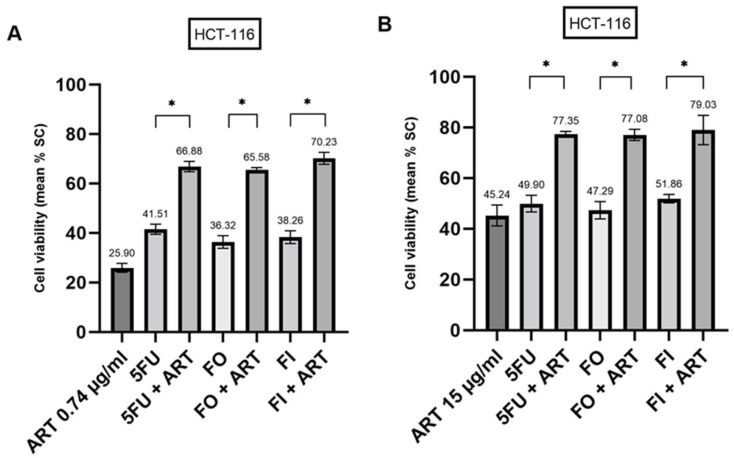
Chemo-sensitization induced by Artesunate (ART) at a PPC of 0.74 µg/mL (**A**) and 15 µg/mL (**B**) in HCT-116 derived cancer spheroids. 5FU, 5-Fluorouracil; FO, 5FU + Oxaliplatin; FI, 5FU + Irinotecan; SC, solvent control. * *p* < 0.001.

**Figure 4 ijms-24-07844-f004:**
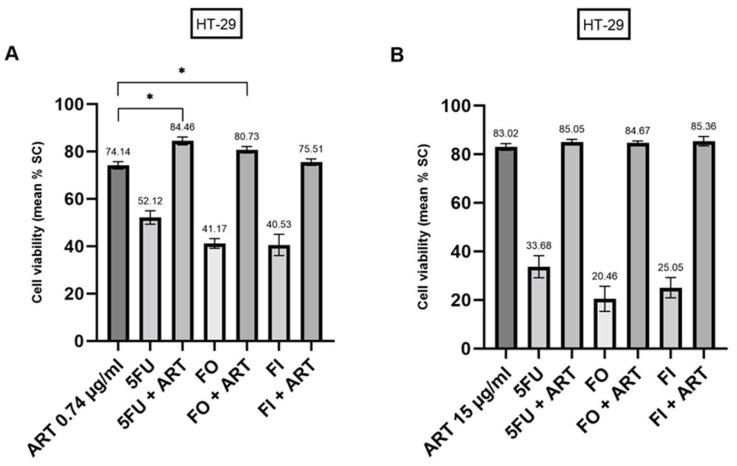
Combination therapy of 5FU, FO and FI with Artesunate (ART) at 0.74 µg/mL (**A**) and 15 µg/mL (**B**) in HT-29 derived cancer spheroids. 5FU, 5-Fluorouracil; FO, 5FU + Oxaliplain; FI, 5FU + Irinotecan; SC, Solvent Control. * *p* < 0.01.

**Figure 5 ijms-24-07844-f005:**
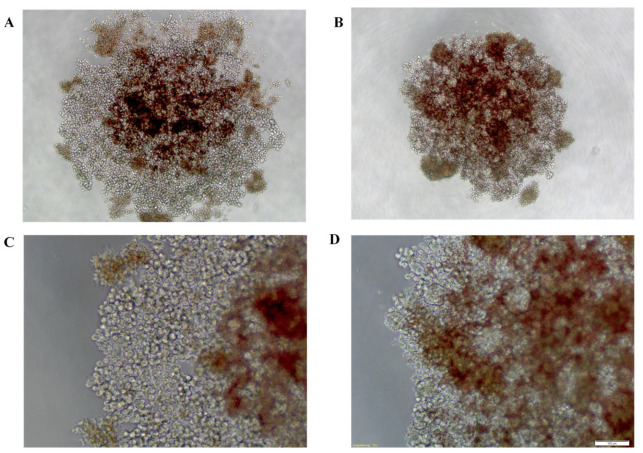
Patient-derived cancer spheroids treated with Solvent Control of Artesunate (**A**,**C**) or 0.74 µg/mL of Artesunate (**B**,**D**); Photographs (**A**,**B**) show spheroids at 40× magnification, photographs (**C**,**D**) show them at 100× magnification; white bar in the corner shows a length of 100 µm.

**Figure 6 ijms-24-07844-f006:**
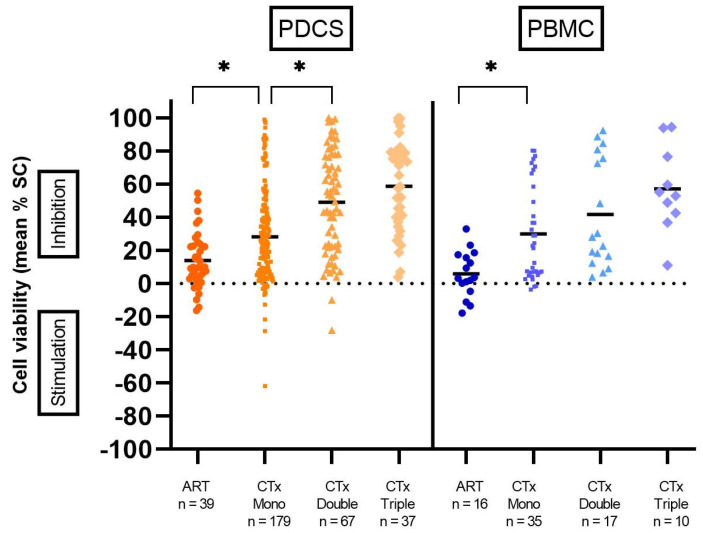
Effect of Artesunate (ART) on patient-derived cancer spheroids (PDCS) and peripheral blood mononuclear cells (PBMC). For reference, chemotherapies (CTx) are shown as single agents (CTx Mono), doublets (CTx Double), and triplets (CTx Triple). * *p* < 0.05; SC, Solvent Control; n, number of tests performed. Bars represent the mean.

**Figure 7 ijms-24-07844-f007:**
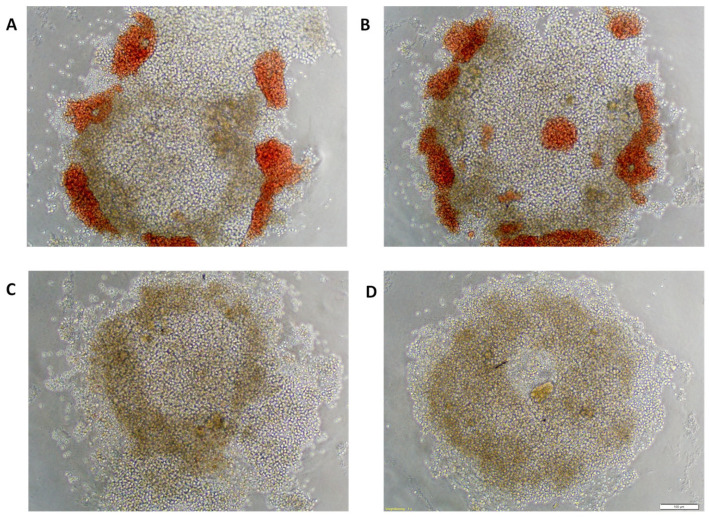
Artesunate (ART) treatment of PBMC induces hemolysis in a dose-dependent manner. (**A**) solvent control corresponding to the ART PPC 15 µg/mL. (**B**) ART PPC 0.74 µg/mL. (**C**) ART PPC 3.26 µg/mL. (**D**) ART PPC 15 µg/mL. Photographs show PBMCs at 100× magnification; white bar in the corner shows a length of 100 µm.

**Figure 8 ijms-24-07844-f008:**
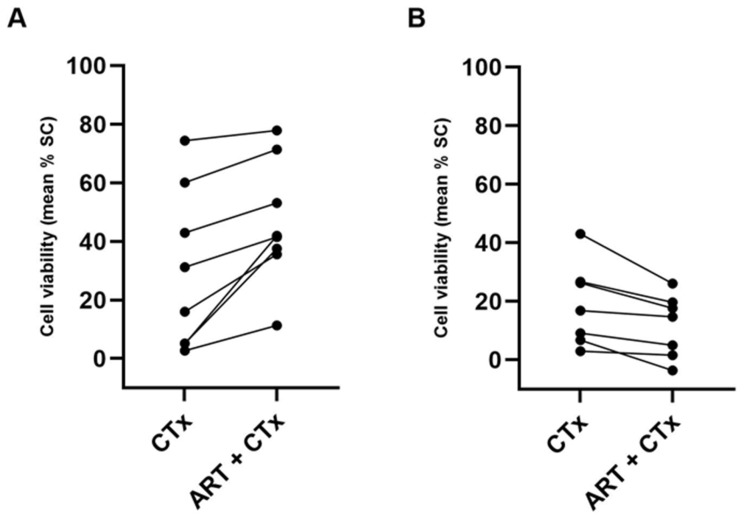
Chemo-stimulating (**A**) and Chemo-reducing effects (**B**) of Artesunate in patient-derived cancer spheroids. ART, Artesunate; CTx, Chemotherapy; SC, Solvent Control.

**Figure 9 ijms-24-07844-f009:**
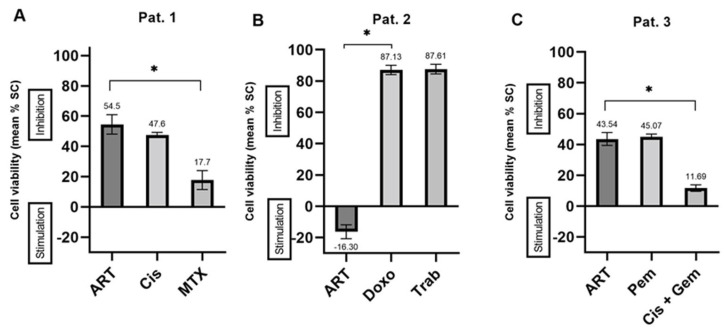
Viability-modulating effects of Artesunate in patient derived cancer spheroids. Pat 1 (**A**), strongest inhibition of cell viability, recurrent cervical cancer. Pat 2 (**B**), strongest stimulation of cell viability, recurrent liposarcoma; Pat 3 (**C**), inhibition of cell viability equally strong as tested cytostatics, primary metastatic breast cancer. ART, Artesunate; Cis, Cisplatin; MTX, Methotrexate; Doxo, Doxorubicin; Trab, Trabectedin; Pem, Pembrolizumab; Gem, Gemcitabine; SC, Solvent Control; * *p* < 0.001.

**Figure 10 ijms-24-07844-f010:**
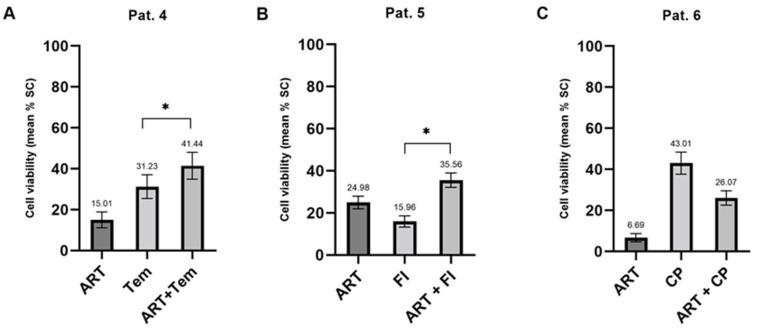
Chemo-modulating effects of Artesunate in patient-derived cancer spheroids. Pat 4 (**A**), chemo-sensitization, recurrent glioblastoma. Pat 5 (**B**), chemo-sensitization, metastatic gastric cancer. Pat 6 (**C**), chemo-reduction, primary ovarian cancer. ART, Artesunate; Tem, Temozolomide; FI, 5-Fluorouracil + Irinotecan; CP, Carboplatin + Paclitaxel; SC, Solvent Control; * *p* < 0.001.

**Table 1 ijms-24-07844-t001:** Patient characteristics. Progressed tumors included primary metastasized tumors with distant metastases and relapsed tumors. N, number of patients.

Parameter		N
Gender	Female	25
	Male	14
Age	<40 years	5
	40–49 years	5
	50–59 years	10
	60–69 years	12
	>70 years	7
Tumor status	Primary non-metastasized	12
	Progressed	27
Tumor entity	Breast cancer	14
	Rare tumors	11
	Prostate cancer	6
	Cervical cancer	2
	Ovarian cancer	1
	Oesophagogastric junctional adenocarcinoma	1
	Gastric cancer	1
	Colorectal cancer	1
	Melanoma	1
	CUP	1

**Table 2 ijms-24-07844-t002:** Concentrations of Artesunate and guideline-recommended cytostatics used in cell-line derived cancer spheroids; PPC, Peak Plasma Concentration.

Treatment	PPC	References
Artesunate	0.74 μg/mL	[68]
	3.26 μg/mL	[69]
	15.00 μg/mL	[70,71]
5-Fluorouracil	100 μg/mL	[72]
Oxaliplatin	3.2 μg/mL	[73]
Irinotecan	7.7 μg/mL	[74]

## Data Availability

The data presented in this study are available on request from the corresponding author.

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
