# Peer review of "Anti-Cancer Effects of Artesunate in Human 3D Tumor Models of Different Complexity"

_ijms, 2023, doi:10.3390/ijms24097844_

Round 1
Reviewer 1 Report
The title of the manuscript is good. English language is in good quality. Figures and Tables are acceptable.The multiple references are the citation problems in this manuscript. The authors should exert some modifications in the part "Results" and "Discussion"
1. Please reform keywords
"cell-line derived cancer spheroid model" and "patient derived cancer spheroid model" are two keywords that seems to be inappropriate
2. All multipple and middle sentences should be reformed in the manuscript
3. Line 44-45 in page 1 seems to disrupts the consistency of the text. Please reconsider this line. It seems to be unnecessary here.
4. Line 46-48 needs proper reference
5. In page 2, part " 2.1.1. Cell line characteristics"
This part belongs to "materials and methods"
Table 1 is also belongs to "materials and methods" not here.
6. About the part "Discussion" in page 9-10
Please categorize all of your findings from the most important to the least inportant one. After that, turn each one of them into subheading and discuss about them one by one
7. Page 10, line 284-285
The sentence "Cell line-derived cancer spheroids were prepared as described previously [66,69,70]." has three references. Why?
8. Line 304-305
Why this line has multipple references?
9. In some parts of "material and method" there are some multipple references for some protocoles. Please explain why you have used multipple references for one protocole.
10. Line 327 and 332 in page 11
Was it not better to draw a table here and insert data in it instead of inserting multipple reference here?
11. About figure 10
Why you have inserted this figure here? Was ir not better to insert this figure and its related data to part "Results"?
12. Please check and adjust the "Reference list" based on the regulations of reference list of journal. (Titles, doi, the name of journal and ... )
Reviewer 2 Report
Authors described anti-cancer effects of anti-malaria drug Artesunate (ALT). This article has partially meaningful data, but must be revised in the following points
major points
The number of data in Figures 7 and 9 is too small. Additional experiments must be done.
Also, the logic of Figures 7 and 9 with Figures 2, 3, and 4 is broken. A clear logical description is needed. In addition, the last two lines of the abstract should be rewritten appropriately.
In the "points to reviewer" when revised, please describe the changes in plain language.
minor point
Table 2 is labeled "data not shown", but this should be clearly described.
Reviewer 3 Report
This article “Anti-cancer Effects of Artesunate in Human 3D Tumor Models of Different Complexity” investigated anti-malaria drug Artesunate (ART) in preclinical 3D cancer models of increasing complexity using clinically relevant peak plasma concentrations to obtain further information for translation into clinical use. The results showed sufficient experimental data, and the content was in line with readers' interests of IJMS. While there were still some shortcomings that need to be further explained or improved.
Comments:
Q1. How would authors define 3D cancer models? Could it be called as 3D cancer models, when the cells are dense enough?
Q2. What are the advantages of this model except strong drug resistance?
Q3. Fig. 1, Fig. 5, Fig. 10. Please add suitable scaleplates.
Q4. Fig. 2 to Fig. 4. How are the dosages of ART selected? Please confirm the p < 0.01 or p < 0.001.
Q5. The results of Fig. 4 were not consistent with the demonstration of Line 365 “Anti-cancer activity of ART as a single agent is low”.
Q6. Why were Oxaliplatin and Irinotecan all combined with 5-FU?
Q7. Did CTx show stronger inhibitory effects on PMBC?
Q8. The selection of these drugs for various cells inhibition is obscure.
Q9. Experimental data involving CTX were not marked for significance.
Round 2
Reviewer 1 Report
All of my comments have been considered. Thank you
Reviewer 2 Report
I think that authors have partially changed their manuscript. Further decisions should be made by the other reviewer and the editor.